# Nanomedicine for Treating Diabetic Retinopathy Vascular Degeneration

Tatiana Borodina [1,2,†], Dmitry Kostyushev [2,3,†], Andrey A. Zamyatnin, Jr. [2,4,5,6] and Alessandro Parodi [2,4,*]

1   Federal Scientific Research Centre "Crystallography and Photonics" of Russian Academy of Sciences, 119333 Moscow, Russia; borodina@crys.ras.ru
2   Scientific Center for Genetics and Life Sciences, Division of Biotechnology, Sirius University of Science and Technology, 354340 Sochi, Russia; dkostushev@gmail.com (D.K.); zamyat@belozersky.msu.ru (A.A.Z.J.)
3   National Medical Research Center of Tuberculosis and Infectious Diseases, Ministry of Health, 127994 Moscow, Russia
4   Institute of Molecular Medicine, Sechenov First Moscow State Medical University, 119991 Moscow, Russia
5   Belozersky Institute of Physico-Chemical Biology, Lomonosov Moscow State University, 119992 Moscow, Russia
6   Faculty of Health and Medical Sciences, University of Surrey, Guildford GU2 7X, UK
*   Correspondence: aparodi.sechenovuniversity@gmail.com; Tel.: +7-9-654-399-009
†   These authors equally contributed to the manuscript.

**Abstract:** The incidence of diabetes and the pathological conditions associated with chronic hyperglycemia is increasing worldwide. Among them, diabetic retinopathy represents a leading cause of vision loss, causing a significant structural and functional impairment of the retinal and choroidal capillary network. Current therapies include anti-angiogenic and anti-inflammatory drugs administered through repetitive and invasive intraocular injections, and associated with significant adverse effects. The presence of ocular barriers affects the efficiency of topically administered therapeutics for treating the posterior segment of the eye. In this scenario, nanomedicine could improve current therapies for diabetic retinopathy by providing tools that can decrease the number of injections thanks to their controlled release properties, while some materials showed a natural ability to mitigate pathological neo-angiogenesis. Moreover, specific surface modifications could open new scenarios for the development of topical treatments. This review describes current advances in generating nanomedicine for diabetic retinopathy, focusing on the properties of the different materials tested explicitly for this purpose.

**Keywords:** diabetic retinopathy; ocular barriers; nanomedicine; polymer nanoparticles; albumin nanoparticles; inorganic nanoparticles; extracellular vesicles

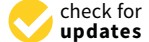

## 1. Introduction

Diabetes mellitus is a group of diseases characterized by chronic hyperglycemia, and the most common expressions of this condition are type 1, type 2, and gestational diabetes [1]. While type 1 diabetes is an autoimmune disease causing an abnormal immune response against insulin-producing β-cells, significantly blunting the expression of this hormone [2], type 2 [3] and gestational diabetes [4] are also characterized by a certain level of skeletal muscle, liver, and adipose tissue insulin resistance. Genetic, environmental, and behavioral factors, as well as maternal obesity, can cause these conditions. Diabetes incidence is increasing worldwide [5]: the population affected by this disease triplicated in the last 40 years [6], and it keeps on growing [7]. Diabetes-associated conditions affect multiple organs and organ systems, resulting in polyneuropathy [8], angiopathy [9], infections [10], nephropathy [10], dementia [11], cardiovascular complications [12], lower limb amputation [13], and blindness [14]. These phenomena are often irreversible and accompanied by a structural and functional impairment of the tissue microcirculation [15]. Vascular degeneration is particularly prominent in the eye, leading to diabetic retinopathy

(DR) [16]. DR affects patients aged between 20 and 65 [17], and its symptoms appear about ten years after diabetes onset [18]. It has been estimated that 20–30 million patients are at risk of irreversible vision loss because of DR [19,20], which is currently one of the leading causes of visual impairment (Figure 1) [21–23] and the leading cause of blindness in preventable retinal diseases [24].

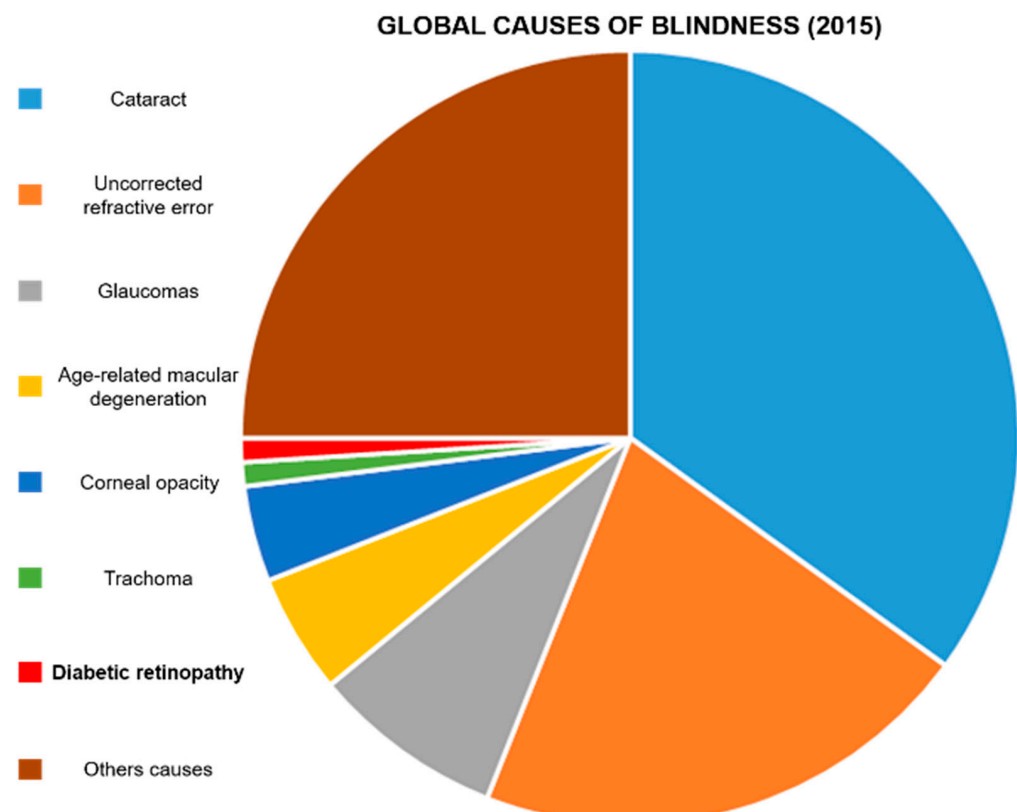

**Figure 1.** Global causes of blindness: In 2015, it was estimated that DR caused blindness in 400,000 people. Diabetic retinopathy represents 1% of the total blind population. Data were extrapolated from Akland et al. [12].

DR is caused by a significant retinal and choroidal capillary network degeneration, due to a local chronic inflammation sustained by advanced-glycation end-products (AGEs) [25], reactive oxygen species (ROS) [26], growth factors, and interleukins [27,28].

In DR, the blood–retinal barrier (BRB) becomes highly permeable, causing local edema, necrosis, and ischemic phenomena [29]. The BRB function depends on the integrity and proper conformation of endothelial intercellular junction complexes that are already significantly affected during the first phases of DR. A high vascular endothelial growth factor (VEGF) pathway activity [30] and a misbalance in ROS generation and elimination [31–33] induce junction disassembly and leaky blood capillary formation [34]. These phenomena can also affect the protein composition of the vitreous humor [35].

To re-establish the normal tissue vascular flow, a neo-angiogenesis process driven by VEGF expression occurs in the retinal and choroidal tissue [36], potentially causing retinal detachment and vision loss [37].

## 2. Clinical Management of DR

Clinically, DR is classified in non-proliferative and proliferative DR, and in six stages of retinal degeneration, varying from small bleeding events to significant neovascularization and retinal detachment (Figure 2) [38–40]. In addition to structural decline, the stress occurring during DR affects photoreceptor cell function and viability [41].

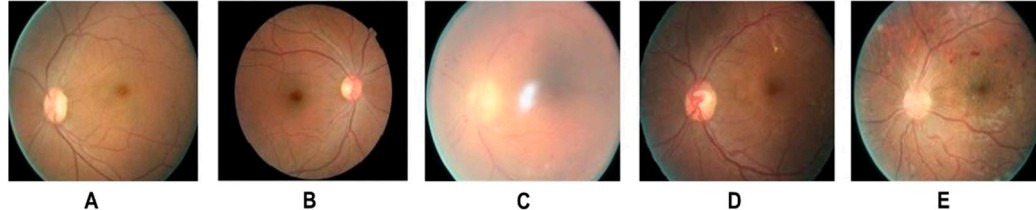

**Figure 2.** Different stages of DR: (**A**) Healthy retina (**B**) mild NPDR (detectable presence of micro aneurysms) (**C**) moderate NPDR (significant presence of micro aneurysms) (**D**) severe NPDR (Intraretinal microvascular abnormalities) (**E**) PDR (presence of hemorrhages). Adapted from Shankar et al. [22].

Optical coherence tomography and fluorescein angiography are commonly used to properly diagnose DR [42], while current therapies for the advanced states of this condition include invasive interventions based on laser photocoagulation (pan-retinal photocoagulation [43]) and vitrectomy [44]. Laser therapy aims at mitigating potential hemorrhagic events by ablating retinal capillary microaneurysms. This procedure requires repetitive applications, and can cause hemorrhages. On the other hand, vitrectomy is a surgical intervention performed to reduce tissue edema and clean the retina from cellular and tissue debris.

Systemic pharmacological treatments [45] aim at lowering blood glucose [46] and lipid [47] levels by adjusting doses and frequency of insulin administration, or administering therapeutics like fenofibrate [48] and statins [49], respectively. In this context also anti-hypertensive therapies [50] showed beneficial properties to mitigate DR progression. However, pharmacological treatments locally administered via subretinal and intravitreal injections are preferred to rapidly and vigorously target the posterior segment of the eye [7].

Typical medications for this disease target the vascular pathologic process and include vascular protective [51] and anti-inflammatory drugs [52,53], angiogenesis inhibitors [54–57], modulators of the microcirculation [58,59]. More recently, peroxisome proliferator-activated receptor (PPAR) agonists showed beneficial effects in reducing inflammation, normalizing vascular function, and mitigating ROS-associated damages [60]. These therapies can have a heterogeneous patient response, side effects, and sometimes high associated costs [49,61–64].

In addition, when locally inoculated, they still need to overcome the ocular biological barriers, significantly mitigating their retinal targeting properties, without mentioning the discomfort provoked to the patient by these therapies.

Nanomedicine can provide different benefits to improve DR treatments, increasing therapeutic residence time in the eye and providing controlled drug release. This review will examine different nanoplatforms tested for this purpose, focusing on the material properties that make these technologies attractive for DR. In particular, we focused on technologies designed to normalize vascular degeneration in diabetic retinopathy that represent the primary cause of this disease. A few attempts [65–67] to develop nanotechnology for reversing and mitigating diabetic retinopthy neurodegeneration were recently performed, but they do not represent the subject of this review. Article inclusion was performed using Google Scholar and PubMed search engines between august and November 2021. A date sorting filter was applied to include papers not older than five years. To select the included articles, we performed literature research using combinations of the following keywords: diabetic retinopathy; ocular barriers; nanomedicine; polymer nanoparticles; albumin nanoparticles; inorganic nanoparticles; extracellular vesicles.

## 3. Ocular Biological Barriers and Investigation Models

The eye structure is very complex and, in the context of its treatment, it is essential to define if the disease affects the anterior or the posterior part of this organ. In the case of diseases affecting the anterior eye, topical treatments can be very effective. On the other hand, invasive intraocular administrations of pharmaceutics are necessary to treat

the tissues of the posterior eye. When administered topically, a drug meant to reach the posterior eye needs to overcome static and dynamic tissue barriers. The static barriers are represented by the tissue layers of the anterior eyes, including the cornea, conjunctiva, sclera, Bruch's membrane, and retinal pigment epithelium [68]. The same retina is a complex tissue composed of different cellular layers that need to be crossed to treat DR and reach its vasculature [69,70]. All the physiological processes that determine the homeostasis of this organ, such as tear fluid, lymphatic vessels, and the conjunctival and choroidal blood flow, instead represent the dynamics barriers. As a result of these barriers, only 5% of the topically administered dose reaches the aqueous humor, and less than this amount can reach the vitreous humor [71]. Finally, the tissues of the anterior segment of the eyes are very sensitive, and when treated with pharmaceutics, they could suffer from dryness, inflammation, and damage, particularly of the cornea [72]. For these reasons, intravitreal injections are often preferred to systemic or topical administration, since they guarantee efficient concentration of the drugs at the disease site. Unfortunately, these treatments usually require multiple injections that, in the course of the therapy, can cause significant complications, including inflammation, retinal damage and detachment, intraocular pressure increase, cataract, and bleeding [73,74], negatively affecting patient compliance [75]. The vitreous humor of the posterior segment is a hydrogel characterized by the presence of more than 50 proteins, with a relative abundance of two main biological components representing 1–2% of this matrix: hyaluronic acid (HA) and collagen [76]. Other proteins are less represented, and besides structural elements, proteolytic enzymes also reside in the ocular environment, and can affect the stability of protein-based biologics [77].

The diffusion of any therapeutic in this matrix is limited and, for this reason, it can be considered as a physical barrier, hampering the retinal targeting. The pioneering work of Patel et al. [78] focused on developing an ex vivo model of vitreous humor to evaluate the diffusion and residence time of protein therapeutics (e.g., anti-angiogenic monoclonal antibodies like bevacizumab and ranibizumab) in the posterior segment. While these therapeutics efficiently decrease pathological neo-angiogenesis, their clearance significantly affects the efficiency of the treatments [79]. Three models of investigation named static, semi-dynamic, and dynamic ex vivo intravitreal models were developed, differing in the number of semipermeable membranes and buffering chambers surrounding the vitreous fluid (isolated from porcine eyes). The best model in investigating the stability of a protein formulation injected in this matrix was represented by the dynamic model, because it could recapitulate the clearance mechanism and pH buffering occurring in this biological environment, while allowing for investigating the diffusion of macromolecules (Figure 3). Even though this work provided a valuable tool for investigation in the field, the authors concluded that the dynamic model still could not recapitulate the complexity of the human eye.

On the other hand, ex vivo models are costly and time-consuming and allow only a few hours of investigation before the tissues irreversibly degenerate [80], while current treatments are supposed to exert their function for days after the injection. Therefore, to generate precise pharmacokinetic data, the use of in vivo models is still paramount. Finally, it is worth mentioning that aging and pathological conditions can significantly change the vitreous humor composition [81]; therefore, standardized in vitro models can only provide vague indications.

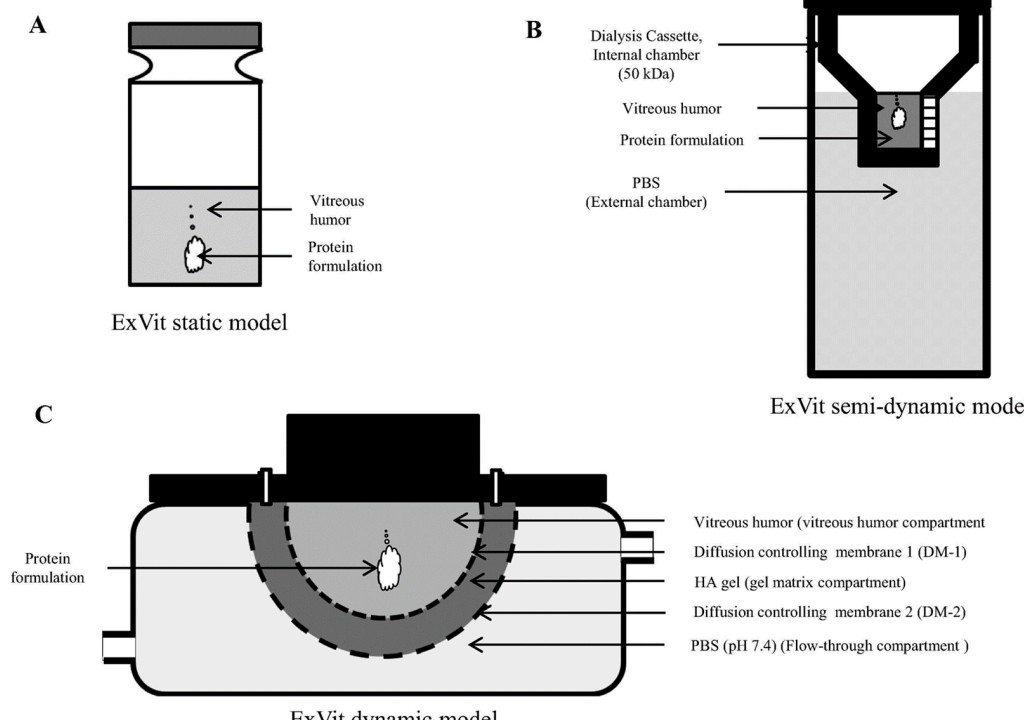

**Figure 3.** Schematic of (**A**) static, (**B**) semi-dynamic and (**C**) dynamic model of vitreous humor diffusion chambers. Reprinted from Patel et al. [51].

## 4. Nanomedicine Application in DR

The application of nanomedicine in DR could improve current therapies for this disease. Drug encapsulation in nanostructure can increase drug solubility and retention in the vitreous humor after the injection. In addition, controlled release properties can reduce the number of injections necessary to achieve significant clinical results. Nanocarrier size, surface charge, and shape are fundamental parameters to consider for developing effective drug formulations for intravitreal injections. While a larger size can increase the drug retention time and controlled release, it can negatively impact particle diffusion in the vitreous humor and retinal targeting. Carriers smaller than 500 nm showed a certain degree of diffusion that was inhibited when the particles were larger than one micron [80].

On the other hand, Koo et al. correlated particle diffusion with their surface charge [70]. They synthesized seven groups of polymeric, hybrid delivery platforms with a relatively narrow size range (between 200 and 350 nm) and very different surface charges (between -25 and +30 mV). The scientist discovered that a strong positive charge inhibited particle (polyethyleneimine-PEI) diffusion in the vitreous humor, probably due to ionic interactions with the negatively charged proteins of this matrix. On the other hand, a moderate positive charge coupled with antifouling agents (glycol chitosan alone or hybridized with PEI) significantly increased their diffusion in the vitreous humor, and allowed retinal inner limiting membrane targeting. Finally, anionic particles based on albumin could also penetrate the deeper layers of the retina. However, the size of these particles was not compatible with the pore size of the retinal tissue, and more investigation indicated that Muller cells could favor this process, likely through active transport. Moreover, the shape of the particles could represent an essential factor in designing retina-targeted nanocarriers. In particular, Shafaie et al. [80] demonstrated that rod-like nanoparticles could efficiently diffuse in the vitreal matrix, but more research on this topic is needed. Different materials and surface properties showed promising results to generate nanomedicine for DR. The following sections will discuss the properties of different delivery platforms shown to mitigate DR degeneration.

### 4.1. Natural and Synthetic Polymers

Polymers represent a heterogenous class of well-characterized materials used to generate nanotherapeutics, thanks to their high encapsulation properties, easy surface modification, and ability to form hybrid technologies with other materials and develop more efficient delivery systems [82]. Polymeric nanoparticles can be formulated from natural (i.e., chitosan) or synthetic molecules (i.e., poly(D, L-lactide-co-glycolide (PLGA))).

Chitosan is a biopolymer derived from the chitin shell of crustaceans and can be easily manipulated in the micron- and nanoscale, generating biodegradable carriers with controlled release properties. Chitosan was tested for the intravitreal delivery of bevacizumab (anti-VEGF antibody) loaded in the particle core or absorbed on their surface to reduce the number of injections associated with this drug [83]. As expected, compared to free injected bevacizumab, similar doses of the encapsulated drug decreased VEGF mRNA expression after two months (Figure 4).

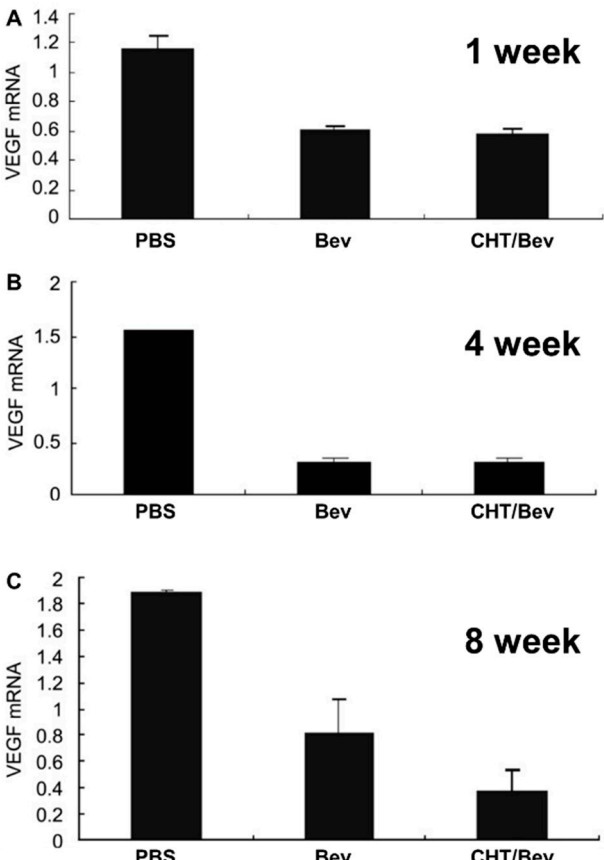

**Figure 4.** Effects of bevacizumab-loaded chitosan nanoparticle on the mRNA expression of VEGF in an in vivo model of DR: VEGF m-RNA expression were evaluated 1 (**A**), 4 (**B**) and 8 (**C**) weeks after intravitreal injection with PBS, non-encapsulated bevacizumab (Bev) and bevacizumab encapsulated in chitosan nanoparticles (CHT/Bev). Reprinted from Lu et al. [55].

Oh et al. [84] described a novel nanoformulation based on HA to deliver Anti-Flt1 peptide to the retina. This seven-amino acid (GNQWFI) peptide can inhibit the interaction between VEGFR1 and its natural ligands (VEGFA, VEGFB, and placental growth factor). In addition, compared to other monoclonal antibodies, it is safer and more cost-effective. HA was shown to regulate the retinal capillary network generation and interact with several receptors expressed in this tissue, like CD44 [84]. For these reasons, it is extensively investigated in the field to improve ophthalmic therapeutics [85]. Spontaneous micelle formation occurred after HA chemical conjugation with the Anti-Flt1 peptide, exploiting the hydrophilic character of the HA and the hydrophobic character of the peptide. The

authors proved the anti-angiogenic properties of this formulation in vitro, in comparison with unconjugated particles, monitoring micelle ability to inhibit VEGFR1/VEGF binding. More importantly, in vivo, functionalized micelles showed an increased ability to mitigate the degeneration of the retinal neo-angiogenesis in a DR rat model. These data were corroborated through pharmacokinetic studies, demonstrating that the micellar formulation prolonged the ocular residence time of the peptide after intravitreal injection.

PLGA represents one of the gold standards of polymer nanomedicine. Qiu et al. encapsulated fenofibrate in PLGA nanoparticles [86] to mitigate DR degeneration. This therapeutic is a PPARα agonist with anti-angiogenic and anti-inflammatory properties [87]. However, its typical oral administration and intravitreal injection are not convenient because of its fast clearance [86]. The encapsulation of fenofibrate in PLGA carriers allowed for a controlled release of the drug, potentially decreasing the number of needed injections.

The scientists demonstrated that high molecular weight PLGA was more efficient in encapsulating the therapeutics and, to increase their dispersity, a polyvinyl alcohol coating was applied. The high molecular weight of the polymer also increased the sustained release of the drug that was still detectable after two months, both in vitro and in vivo. Compared to empty particles, the efficacy of fenofibrate-loaded polymeric carriers improved retinal function, decreased edema formation, leukostasis, and inflammatory cytokine secretion, with no evident adverse effects in a model of streptozotocin (STZ)-induced diabetic rats. A similar technology has been developed to deliver the PPARγ agonist pioglitazone. In this case, the PLGA nanoparticles were coated with polysorbate 80 to increase the interaction with the eye surface and enhance drug delivery via topical treatment [88]. Different ratios between lactide and glycolide (75:25 and 50:50) were tested for drug encapsulation efficiency and release. The particles showed similar loading efficiency, but the 75:25 formulation allowed for a prolonged pioglitazone release, even though the 50:50 formulation resulted more efficient in decreasing VEGF levels in the vitreous humor of STZ-induced diabetic rats. The authors attributed this effect to particle relative higher burst release and ability to negotiate the ocular barriers, even though more investigations are necessary to understand their trafficking.

Polycaprolactone nanoparticles coated with Pluronic F68 (PF68) were also tested to achieve retinal targeting via topical administration. PF68 penetrating mucoadhesive properties allowed the particles to cross the external ocular barriers, while the encapsulation in polycaprolactone increased drug release [89]. The system was tested to deliver the corticosteroid triamcinolone acetonide [24], a therapeutic well-known in clinics for its anti-inflammatory and anti-angiogenic properties, but like many other therapeutics, its usage for DR needs repetitive injections. Compared to free drug administration and empty nanoparticles, loaded nanoparticles normalized the retinal tissue after 40 days of topical treatment, and decreased inflammatory molecule expression and pathological angiogenesis.

### 4.2. Albumin Nanoparticles

Proteins represent a very investigated material for nanoparticle synthesis, due to their biocompatibility, physiological degradation, and efficient surface chemical modification. Albumin is perhaps the most studied protein in the field, since it is cost-effective, has some natural properties to binding therapeutics, and can be formulated in nanostructures with different protocols [90]. Finally, albumin can be easily hybridized with other materials like polymers, increasing particle drug encapsulation and trafficking through various biological barriers [90].

Albumin nanoparticles generated via desolvation followed by crosslinking method were tested to improve the ocular delivery of apatinib. This therapeutic is a VEGFR2 inhibitor effective for DR treatment via intravitreal injection. Apatinib showed promising results in inhibiting pathological neo-angiogenesis, but its hydrophobic character has been associated with adverse effects. For this reason, encapsulation was used to increase its solubility and, eventually, its safety [71]. The particles were also coated with HA to increase

their interaction with mucins. In this context, HA coating increased particle ability to diffuse in the vitreous humor and favored their internalization in retinal cells expressing CD44. In vitro, the particles showed no toxicity and good ability to interact with mucins. In vivo, they were topically applied or intravitreally injected. In both cases, the loaded particles coated with HA showed higher anti-angiogenic properties compared to loaded, not coated particles. These results were proven via fluorescent microscopy, showing higher retention of the coated particles in the retina.

Interestingly, particle retention for both groups was only observed in the disease model and not in healthy mice, confirming that retinopathy can represent a targetable condition via HA. Similarly, Ji Hoon et al. coated albumin nanoparticles with PEG instead of HA, to increase their hydrophilicity and biocompatibility [91]. This system loaded with apatinib showed an anti-VEGF action in vitro via transepithelial electric resistance measurement. In vivo, intravitreally injected loaded particles inhibited edema generation in a model of DR. The group of Dr. Rupenthal tested the permeation of albumin nanoparticles in the vitreous humor under ultrasound pulse [92] (Figure 5). This effect was due to thermal and physical effects (e.g., cavitation, acoustic, streaming, and mechanical stress). The particles were loaded with an anti-angiogenic peptide derived from Connexin43. The transscleral and, above all, the intravitreal application of ultrasounds significantly enhanced the ex vivo diffusion of the particles in the vitreous humor of bovine eyes, allowing efficient interactions with the deepest layers of the retina. However, the scientists concluded that the transscleral application provided higher benefits, since it was significantly safer and less invasive.

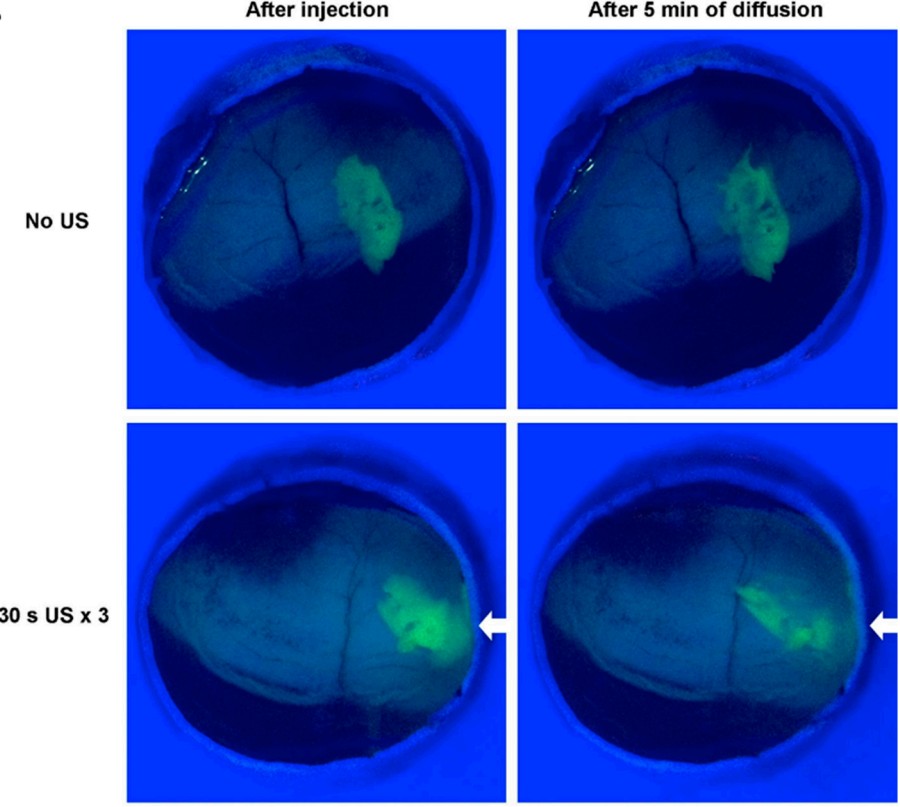

**Figure 5.** Effects of ultrasound pulses on albumin nanoparticles diffusion in vitreous humor. Reprinted from Huang et al. [92].

*4.3. Inorganic Nanoparticles*

Inorganic nanoparticles provide several advantages in formulating delivery carriers, including stability in biological environments and standardized synthesis protocols. Some materials have anti-angiogenic and anti-inflammatory properties, making them perfect

candidates for DR treatment. However, their application in nanomedicine has always been mitigated by concerns about biocompatibility and degradation product safety. Silica has been extensively investigated to generate nanocarriers with proven properties in encapsulating and increasing hydrophobic therapeutics' solubility. In addition, silica nanoparticle biodegradation can be tailored by adjusting their porosity. Despite some safety concerns at the cellular level related to their tendency to aggregate and generate ROS upon prolonged exposition, their use as a food and an oral drug additive is FDA-approved [93]. The group of Dr. Park tested their toxicity after topical ocular administration, in a direct comparison with the same doses of ingested particles [94]. The scientists did not detect any systemic or organ side effects connected to the topic treatment following a 12 week observation period after particle administration. Intravitreally injected silica nanoparticles showed anti-angiogenic effects inhibiting VEGFR phosphorylation after VEGF binding [95], while in another study, they showed promising results for the ocular delivery of tacrolimus [96].

Silicon nanoparticles were tested to develop a theragnostic system to detect pathological neo-angiogenesis in the posterior ocular segment, while providing significant anti-angiogenic effects [97]. The system consisted of ultrasmall silicon nanoparticles functionalized with the anti-angiogenic peptide RGD. The particles demonstrated anti-angiogenic properties in vitro and in vivo after intravenous injection in a murine model of ocular angiogenesis (Figure 6).

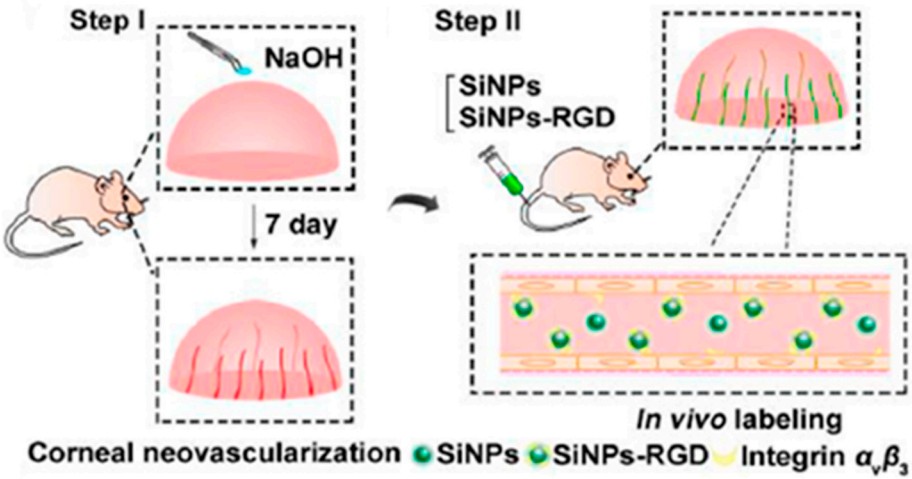

**Figure 6.** Binding ability of SiNPs/SiNPs-RGD to angiogenic blood vessels in the cornea of mice. Schematic diagram of the experimental procedure. Reprinted with permission from [68] Copyright 2018 American Chemical Society.

Gold was shown to have antioxidant and anti-angiogenic properties, probably due to its plasmonic properties [98]. In addition, it demonstrated inhibitory properties against retinal epithelium proliferation in response to VEGF or inflammatory cytokines [99]. On the other hand, Shen et al. [100] related gold nanoparticle anti-angiogenic properties to their ability to induce autophagy affecting cell proliferation, as shown in vitro and in a model of oxygen-induced retinopathy in vivo. Dong et al. [101] fabricated gold nanoparticles via a green-chemistry synthetic process using resveratrol as a stabilizing and reducing agent. The particles resulted uniformly coated with this drug that has well-known anti-inflammatory and anti-oxidative properties. After inducing the disease in rats via STZ injection, the animals were orally administered every day for three months with gold nanoparticles or calcium dobesilate, known to contrast the development of the disease. Gold nanoparticles coated with resveratrol showed similar ability to calcium dobesilate in normalizing retinal capillaries and mitigating vascular leaking. These properties were associated with the retinal expression increase of pigment epithelium-derived factor and decrease of VEGF-1. In addition, the particles were shown to interfere with the inflamma-

tory pathway of the extracellular signal-regulated kinase (ERK) 1/2 -NF-κB p65 activated by STZ, while decreasing the retinal mRNA expression of inflammatory cytokines like TNFα, MCP-1, ICAM-1, IL-6, and IL-1β, even though the mechanism of action was not investigated. Apaolaza et al. described the application of gold nanoparticles conjugated with low molecular weight HA to inhibit AGEs formation and vascular proliferation, as demonstrated in vitro, ex vivo, and in a chorioallantoic membrane assay [102]. Moreover, in this case, HA surface functionalization provided a better targeting against retinal cells expressing CD44, but it was also functional in improving the chemical-physical properties of the system. The HA coating decreased aggregate formation in biological fluids, increased particle mobility in the vitreous humor, and retinal targeting. A nanoformulation of gold nanoparticles loaded with sorafenib was successfully tested to mitigate retinopathy in a disease model induced by DL-α-aminoadipic acid injection [103]. The particles were coated with folic acid to increase their targeting and biocompatibility properties. The group of Dr. Gurunathan demonstrated that silver nanoparticles also possess anti-angiogenic properties. In particular, the particles inhibited in vitro retinal endothelial permeability induced by VEGF and IL-β affecting Src kinase pathway [104].

Commercial iron oxide nanoparticles were loaded with the somatostatin analog octreotide to treat DR [105]. Octreotide is a peptide with proven nerve-protective (anti-apoptotic) and anti-angiogenic properties interfering with the VEGF pathway, but showed gastrointestinal toxicity when orally administered. In this study, octreotide was loaded via surface functionalization, and this nanoformulation demonstrated anti-angiogenic properties in vitro. Interestingly, when the therapeutic was conjugated with the particles, it showed higher anti-angiogenic effects than when freely administered to the cells, probably because drug conjugation increased its stability without affecting its working mechanism. In vivo, the particles demonstrated high biocompatibility and localization in the retina, showing high protective power in explanted retinas treated with hydroxyl peroxide.

### 4.4. Extracellular Vesicles for RNA Delivery

Extracellular vesicles (EVs) are membrane-enclosed lipid bilayer nanoparticles released from essentially every mammalian/human cell [106]. Their size ranges from 30 to 150 nm, and they carry different sets of proteins, lipids, RNA, and DNA molecules (molecular cargo). Naturally, EVs participate in intercellular communication within the body, regulating various processes, such as tissue regeneration, cell migration, and proliferation. EV biological properties and functions are mostly related to the properties of the parental cell line, including phenotypic effects on the recipient cells and targeting. EVs have been extensively studied as delivery vehicles for biologics, due to their high biocompatibility, safety, low immunogenicity, trafficking across biological barriers, and ability to protect their biological payload in vivo [107]. EVs can be manufactured in large-scale exploiting methods based on cell factories or bioreactors [108]. EV loading follows two principal approaches: endogenous and exogenous (post-production) packaging. In the endogenous approach, the EV-producing cell line is induced to overexpress the biological therapeutic, while the exogenous loading occurs through physical or physicochemical methods, like electroporation, generation of hybrid nanosomes, and extrusion following EV isolation and purification [107].

In DR, EVs have been preferentially used to deliver short RNAs, such as small interfering RNAs (siRNAs), microRNAs (miRNAs), and antisense oligonucleotides. Different RNAs can be loaded into EVs using RNA overexpression or by exploiting RNA-binding motifs (i.e., MS2 coat protein) [109] fused to constitutive EV markers (CD9 [110], CD63 [111], CD81 [112], LAMP2B [113], PTGFRN [114], and MFGE8 [115]). Some of these procedures are based on RNA–EV interacting motif engineering [106,116] and EV "zipcode" sequences exploiting specific RNA-binding proteins [117]. miRNAs enrichment substantially occurs via pre-miR-451 backbone insertion that is highly expressed in EVs [118]. Other approaches are based on EV fusion with RNA-loaded liposomes or extrusion methods, but typically at the cost of EV stability and biodistribution properties [107].

DR progression is accompanied by the downregulation of numerous miRNAs, including miRNA-20a-3p, miRNA-20a-5p, miRNA-106a-5p, and miRNA-20b [119]. These alterations can result in abnormal ultrastructural retinal modifications, increased VEGF expression, and reduced brain-derived neurotrophic factor levels, culminating in retinal microaneurysm onset [120]. EVs containing these miRNAs can revert the effects of high-glucose retinal damage when administered systemically or locally, and inhibit detrimental angiogenesis implicated in DR progression [121]. However, compared to local administrations, EV systemic injections are typically less effective [122].

EVs loaded with miRNA-20a-3p, miRNA-20a-5p, miRNA-106a-5p, and miRNA-222 showed significant clinical benefits in vivo against DR experimental models [122–124]. EVs enriched in cPWWP2A, with high affinity for miRNA-579, and normalized retinal microvascular function [125], while miRNA-202-5p-loaded EVs inhibited cell proliferation, migration, and vascularization [126]. In rats, EVs from adipose-derived MSCs loaded with miRNA-192 relieved DR-associated inflammation and pathologic angiogenesis [127]. DR retinal degeneration was improved via EV delivery of miRNA-21 [128], since its deficiency aggravates photoreceptor injury, apoptosis, and retinal degeneration. Adipose stem cell-derived EVs mitigated DR degeneration in diabetic rabbits. It was demonstrated that EVs carrying miRNA-222 suppressed abnormal growth of blood vessels and restored physiological angiogenesis in experimental diabetic in vivo models [122,127]. Similarly, EVs loaded with miRNA-126 overexpressed in umbilical cord blood-derived MSCs successfully inhibited hyperglycemia-induced inflammation and alleviated DR progression in vivo [129]. Bone marrow MSC-derived EVs also inhibited oxidative stress, inflammation, and apoptosis, and promoted Muller cell proliferation. These effects were attributed to the miRNA-486-3p, suppressing the pro-inflammatory TLR4/NF-kB pathway [121].

While some miRNAs can provide clinical benefits to DR, others like miRNA-15a can promote diabetic complications [130]. In DR, plasma-derived EVs [131] and platelet-rich plasma EVs [132] affected retinal microvasculature and endothelial cells, respectively. These data suggest that EV sources should be carefully considered, since harmful miRNAs could be co-delivered with therapeutic biologics.

In addition to miRNAs, long non-coding RNAs (lncRNAs) have also been demonstrated to alter DR progression via diverse mechanisms. MSC-derived EVs enriched in lncRNA SNHG7 inhibited DR progression by "sponging" miRNA-34a-5p and inducing X-box binding protein 1 (XBP1) axis activity, which mitigates the ER stress [133,134]. In parallel, small nucleolar RNA host gene 7 (SNHG7) inhibited DR cell viability, migration, and angiogenesis [135]. EV loaded with miRNA-34a-5p could modulate both SNHG7 and XBP1 expression, interfering with DR progression [133]. NFIA/NLRP3 pathway is involved in DR microvascular complications and endothelial dysfunction, and can represent a molecular target for treating this disease [136]. Delivery of circular RNA Ehmt1 via hypoxia-conditioned pericyte-derived EVs upregulated NFIA expression and suppressed NLRP3-mediated inflammation [137].

New technologies enabling the successful sorting of specific miRNAs (or cohorts of miRNAs) into EVs are yet to be applied for generating DR treatments, considering different vesicle sources, therapeutic effects, and multiple administration routes (Figure 7). To date, a plethora of highly effective approaches are available, but have not been utilized for this specific purpose. Optimizing miRNA release from EVs and increasing their cytosolic concentrations upon local injection are essential tasks to increasing EV application in DR.

The exciting perspectives of EVs for the DR treatment are yet to be defined, as only a concise list of experimental studies related to EV-mediated reversion of DR is currently published.

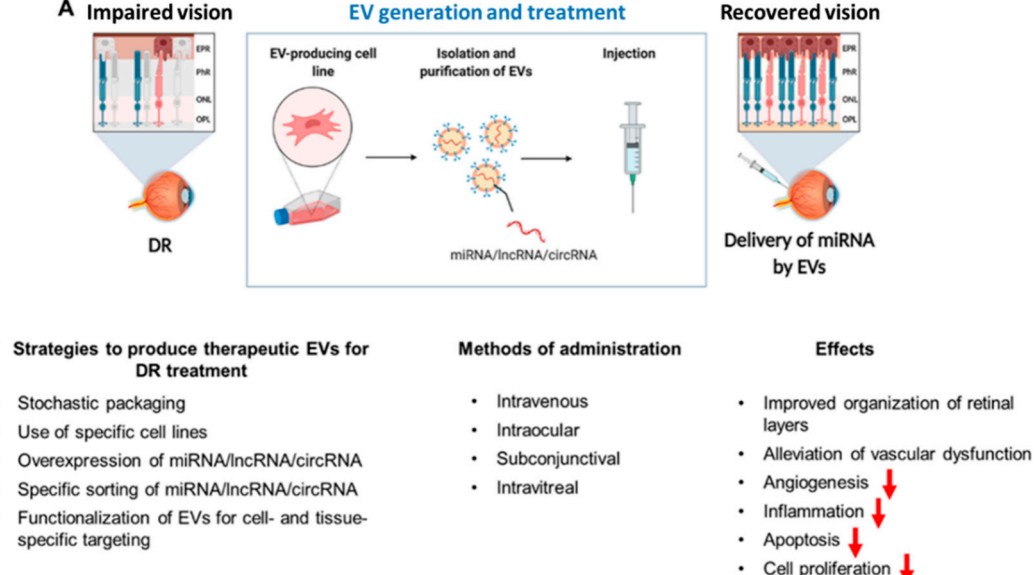

**Figure 7.** Application of EVs for treating DR. (**A**) Pipeline for using EVs for DR treatment. EVs are produced in appro-priate cell lines, isolated, and purified using high-throughput methods. EVs are standardized for their payload (thera-peutic miRNA/lncRNA/circRNA) and used for treating DR. (**B**) Strategies for using EVs in DR and their major therapeutic benefits. Red arrows indicate decrease of the phenomenon.

## 5. Conclusions

DR is a common complication of diabetes, and it will be responsible for vision loss in millions of patients in the coming decades. Current therapeutic approaches are highly invasive and characterized by significant side effects. The application of nanomedicine to optimize pharmaceutical formulations could reduce the number of injections needed to treat this disease by prolonging the drug residence time in the eye and improving drug pharmacokinetic properties. In addition, nanocarriers could expand the portfolio of DR therapeutics by increasing the efficiency of biologics, particularly proteins and RNA molecules. More investigation is also needed in understanding the natural properties of certain materials used to fabricate nanoparticles in mitigating the neo-angiogenesis and the inflammatory process. Finally, some of the studies proposed here registered beneficial effects against DR via the topical or systemic administration of nanoformulations. These data could represent a real breakthrough in the clinical management of DR, but more studies are needed to understand nanoparticle trafficking through ocular biological barriers.

**Author Contributions:** Conceptualization, T.B., D.K., A.P.; writing, T.B., D.K.; review and editing, A.A.Z.J.; supervision, A.P. All authors have read and agreed to the published version of the manuscript.

**Funding:** This research was funded by the Russian Science Foundation (grant # 21-75-30020).

**Conflicts of Interest:** The authors declare no conflict of interest.

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
