# Peer review of "Nanomedicine for Treating Diabetic Retinopathy Vascular Degeneration"

_2673-8937, doi:10.3390/ijtm1030018_

Round 1

Reviewer 1 Report

The authors presented an interesting review on nanomedicine for diabetic rethinopathy. The manuscript is with merit and the findings are worth reporting. However, before publication could be considered, the authors should revise extensively the manuscript and address several concerns. The organization of the manuscript should be improved and the the authors should also revise and improve the language and syntax of the entire manuscript.

Line 1-2: the very first sentence of the manuscript [“Diabetes is a pathology due to different causes (Type 1, Type 2, gestational, ...) and is characterized by chronic hyperglycemia.”] should be edited and the authors should be very precise in the definitions of the various causes of diabetes with corresponding references.

Entire manuscript: the authors should avoid list followed by “…” as they did repeatedly in the manuscript: for example at line 1 (“(Type 1, Type 2, gestational, ...)”), at line 40 (“against infections ....”,): at line 105 “(VEGF-A, VEGF-B, ...)”, 106 “(VEGFR1, VEGFR2, ..) ; line 212: “PEI, ... “. The authors should be very precise in their statements and when describing list it should never be concluded by “…” : please revise the entire manuscript correspondingly.

The authors should provide references for their  statements (for example all the statement from line 37 to 45 are not justified by corresponding references and overall references to statements are missing; statements at lines 121-126 do not present references neither): please revise all the manuscript correspondingly.

The authors should provide some statements to connect the first part of the introduction (line 33-45) to the second one related to DR.

DR: the authors should move the statements about the epidemiology of DR (lines 72-28) at the beginning  (line 46) in order to provide its epidemiology before the discussion of its pathophysiology.

The authors should re-organize the manuscript in order to clarify early in the manuscript the aim of the review. Currently, the aim of the review is indicated for the first time at line 200: it would be important to re-organize the first part of the manuscript (sections 1 to 4) in order for the reader to have a better understanding from the beginning of the scope of the manuscript. It would be important to reduce the length of the first 3 sections and instead give more emphasis to the specific focus of the review. All the subsection of part 4 represent the main core of the manuscript: the authors should emphasize this part of the manuscript by reducing the length of the previous introductory sections in order to give more emphasis on the specific topic of their review, as reflected by the title itself of the manuscript.

A “material and methods” section is missing: the authors should provide one, indicating precisely how the review was conducted (which databases were used, with which timeframe, and indicate specifically the keywords used)

Author Response

The authors presented an interesting review on nanomedicine for diabetic rethinopathy. The manuscript is with merit and the findings are worth reporting. However, before publication could be considered, the authors should revise extensively the manuscript and address several concerns. The organization of the manuscript should be improved and the the authors should also revise and improve the language and syntax of the entire manuscript.

-We thank the reviewer for her/his time and efforts dedicated to improving our manuscript. All the concerns were addressed, and we improved the paper readability.

Line 1-2: the very first sentence of the manuscript [“Diabetes is a pathology due to different causes (Type 1, Type 2, gestational, ...) and is characterized by chronic hyperglycemia.”] should be edited and the authors should be very precise in the definitions of the various causes of diabetes with corresponding references.

-A better definition regarding diabetes causes and new references were included.

Entire manuscript: the authors should avoid list followed by “…” as they did repeatedly in the manuscript: for example at line 1 (“(Type 1, Type 2, gestational, ...)”), at line 40 (“against infections ....”,): at line 105 “(VEGF-A, VEGF-B, ...)”, 106 “(VEGFR1, VEGFR2, ..) ; line 212: “PEI, ... “. The authors should be very precise in their statements and when describing list it should never be concluded by “…” : please revise the entire manuscript correspondingly.

-Generic lists were modified to be more precise.

The authors should provide references for their  statements (for example all the statement from line 37 to 45 are not justified by corresponding references and overall references to statements are missing; statements at lines 121-126 do not present references neither): please revise all the manuscript correspondingly.

-More references were included in particular (but not exclusively) in the first paragraphs of the manuscript.

The authors should provide some statements to connect the first part of the introduction (line 33-45) to the second one related to DR.

-A connecting paragraph was included

DR: the authors should move the statements about the epidemiology of DR (lines 72-28) at the beginning  (line 46) in order to provide its epidemiology before the discussion of its pathophysiology.

-We moved this part at the beginning of the introduction

The authors should re-organize the manuscript in order to clarify early in the manuscript the aim of the review. Currently, the aim of the review is indicated for the first time at line 200: it would be important to re-organize the first part of the manuscript (sections 1 to 4) in order for the reader to have a better understanding from the beginning of the scope of the manuscript. It would be important to reduce the length of the first 3 sections and instead give more emphasis to the specific focus of the review. All the subsection of part 4 represent the main core of the manuscript: the authors should emphasize this part of the manuscript by reducing the length of the previous introductory sections in order to give more emphasis on the specific topic of their review, as reflected by the title itself of the manuscript.

-We reorganized and shortened the intro. Please consider that reviewer 2 asked for including more information. 

A “material and methods” section is missing: the authors should provide one, indicating precisely how the review was conducted (which databases were used, with which timeframe, and indicate specifically the keywords used)

-A material and method section was included indicating the used databases, while the keyword section was implemented. However, we are not sure if a review should contain such a section. We leave the editor the last decision.

Reviewer 2 Report

This paper reviews current DR treatments with focus on small molecule and EV vehicles. There are a number of reviews on this topic, but overall, this paper is well-written and a good introduction to the subject. Suggestions are made to improve the quality of the manuscript.

-Mention of systemic treatments for DM that also influence DR progression, including therapeutics to help manage glucose levels, should be added (Section 2. Clinical management).

-line 58- 'BRB is highly permeable...' Please edit for accuracy, increased permeability is different than highly permeable and also relative to non-diabetic conditions.

-Clarification regarding the major therapeutic objectives should be clearly stated for all mentioned treatments. For example, to decrease VEGF levels, anti-inflammatory, etc.

-Please remove the use of (...) throughout the manuscript and correct typographical errors.

Author Response

This paper reviews current DR treatments with focus on small molecule and EV vehicles. There are a number of reviews on this topic, but overall, this paper is well-written and a good introduction to the subject. Suggestions are made to improve the quality of the manuscript.

-We thank the reviewer for her/his time and efforts dedicated to improving our manuscript quality. All the concerns were addressed, and we improved the paper readability.

-Mention of systemic treatments for DM that also influence DR progression, including therapeutics to help manage glucose levels, should be added (Section 2. Clinical management).

-A short paragraph mentioning systemic treatments was included. Please consider that reviewer 1 asked for shortening this section.

-line 58- 'BRB is highly permeable...' Please edit for accuracy, increased permeability is different than highly permeable and also relative to non-diabetic conditions.

-This specific sentence was edited.

-Clarification regarding the major therapeutic objectives should be clearly stated for all mentioned treatments. For example, to decrease VEGF levels, anti-inflammatory, etc.

-More info about the drug targeting was included.

-Please remove the use of (...) throughout the manuscript and correct typographical errors.

-Generic lists were edited to be more precise.

Round 2

Reviewer 1 Report

The authors addressed all the comments. However, as previously indicated I would suggest to give more emphasis and clarify to the reader the aim of the review early on in the manuscript. In addition,  the "material and methods" section was not included as part of the manuscript but only at the end as the following statement: 

"Materials and methods: Article inclusion was performed using Google Scholar and Pubmed search engines between august and November 2021."

This section must be expanded in order to include the specific keywords that the authors used to do the literature search and included in the main manuscript, possibly after a clear statement indicating the aim of the review.

Author Response

R-The authors addressed all the comments. However, as previously indicated I would suggest to give more emphasis and clarify to the reader the aim of the review early on in the manuscript. 

A- We thank again the reviewer for her/his valuable comments. We tries to move the focus of the review to the very first section of the work, but the result was not satisfying. However, we believe a big improvement was done. In the first version of the paper, the focus of the work was introduced at the beginning of section 4. Currently, it is located at the end of section 2, after a brief introduction to the disease and its clinical management.  We believe the word nanomedicine should appear after the description of the pharmacological approaches used to treat this disease because nanomedicine IS a tool to improve drug PK properties.

R-In addition,  the "material and methods" section was not included as part of the manuscript but only at the end as the following statement: 

"Materials and methods: Article inclusion was performed using Google Scholar and Pubmed search engines between august and November 2021."

This section must be expanded in order to include the specific keywords that the authors used to do the literature search and included in the main manuscript, possibly after a clear statement indicating the aim of the review.

A- We included the M&M section where it was indicated in the journal format for the authors. We will discuss with the editor the possibility to include it somewhere in the paper. This part was expanded as requested.